# Efficacy of Iopamidol for Sealing an Injured Thoracic Duct: Pilot Experiments in a Large Animal

**Hyo Yeong Ahn** [1,†], **Seunghwan Song** [1,†], **Up Huh** [1,*], **Il Jae Wang** [2], **Jung Seop Eom** [3] **and Dong-Man Ryu** [4]

1 Department of Thoracic and Cardiovascular Surgery, Pusan National University School of Medicine, Biomedical Research Institute, Pusan National University Hospital, Busan 49241, Korea; doctorahn02@hanmail.net (H.Y.A.); song77.sh@gmail.com (S.S.)

2 Department of Emergency Medicine, Pusan National University School of Medicine, Biomedical Research Institute, Pusan National University Hospital, Busan 49241, Korea; jrmr9933@naver.com

3 Division of Respiratory, Department of Internal Medicine, Pusan National University School of Medicine, Biomedical Research Institute, Pusan National University Hospital, Busan 49241, Korea; ejspulm@gmail.com

4 Biomedical Research Institute, Pusan National University Hospital, Busan 49241, Korea; babopanda@pusan.ac.kr

\* Correspondence: tymfoo82@gmail.com; Tel.: +82-51-240-7267

† Hyo Yeong Ahn and Seunghwan Song contributed equally to this study.

**Abstract:** Chylothorax can be spontaneously healed by lymphangiography using lipiodol, but pulmonary or systemic embolization is a potential complication. We determined the efficacy of iopamidol for treating chylous leakage in an animal model. Twelve pigs were randomly divided into two groups. After inducing thoracic duct damage, pigs from groups A and B were injected with iopamidol and lipiodol, respectively. At 5, 10, and 30 min after damage induction, the drug effects were monitored by video-assisted thoracoscopy and lymphangiography. In vitro, chyle samples from the pigs were incubated with iopamidol and lipiodol. The damaged thoracic duct was harvested and examined using microscopy. In group A, four and two pigs did not show chylous leakage after 5 and 10 min, respectively. In group B, none showed chylous leakage after 5 min. Nevertheless, the *p* value was 0.46, and there was no statistically significant difference between groups A and B. In vitro, both iopamidol- and lipiodol-treated chyle samples adhered after 5 min and solidified at 30 min. Our findings confirmed that the damaged thoracic duct was clogged with an amorphous proteinaceous material (iopamidol). Therefore, iopamidol is potentially a new therapeutic agent for chylous leakage. Thoracic duct embolization failures or systemic embolization risks from lipiodol injection may be avoided by injecting iopamidol via the thoracic duct, and this warrants further investigation.

**Keywords:** chylothorax; thoracic duct embolization; iopamidol

## 1. Introduction

Chylothorax can result from thoracic duct or large lymphatic vessel damage during pulmonary resection or cardiovascular surgery [1–3]. Current treatment options for chylothorax include conservative, surgical, and the more recent interventional radiological procedures [2–11]. Percutaneous transabdominal thoracic duct embolization (TDE), an interventional radiological procedure, has been widely performed to promote early recovery from chylothorax [11]. Although TDE fails in 30% of patients because of thoracic duct variation [9,11–14], chylothorax has shown spontaneous healing immediately after lymphangiography alone, which may affect the healing of chylous leakage [11].

Clinically, lymphangiography has been performed using lipiodol. Lipiodol-induced complications, including pulmonary embolization and cerebral embolization, have been reported [15–17]. To avoid

the serious adverse effects associated with lipiodol, we considered iopamidol (Fuji Pharma, Tokyo, Japan), an iodinated contrast agent used for angiography, to treat chylous leakage. As iopamidol is water-soluble and widely used as an angiographic contrast agent [18], it is expected to have considerably fewer adverse effects from embolization than lipiodol does. If iopamidol and lipiodol have the same therapeutic effect on chylous leakage, it would thus be safer to treat chylous leakage with iopamidol. We conducted animal experiments as a pilot study to confirm the therapeutic effect of both drugs on chylous leakage. We chose pigs as the experimental animal model because they are similar to humans in terms of body size, anatomical features, physiology, pathophysiological reactions, and diet consumption [12]. Pigs are often used to develop and improve biomedical procedures and medical equipment [2].

We present the following article in accordance with the ARRIVE reporting checklist.

## 2. Materials and Methods

### 2.1. Ethical Statement

Experiments were performed under a project license (PNUYH-2019-071) granted by the Institutional Animal Care and Use Committee (IACUC) of Pusan National University Hospital.

### 2.2. Experimental Animal

Twelve healthy white Yorkshire female pigs were used in the study after acclimation for 7 days. The mean age and weight of the pigs were 11.2 ± 1.2 (range, 10–13) weeks and 39.3 ± 1 (range, 38.4–41.1) kg, respectively.

### 2.3. Housing and Husbandry

The breeding facility for the pigs was equipped with a specific-pathogen-free system. We maintained one pig per open cage; for bedding, we used autoclaved pulp chips and shredded aspen. The pigs were fed Teklad 2018S (Envigo, Indianapolis, IN, USA) and provided with sterilized tap water. The management and operation of the lab were in accordance with The Guide for the Care and Use of Laboratory Animals (Eighth Edition). The pigs were maintained under the following conditions: 12/12-h day/night cycle, 21 ± 2 °C temperature, and 60% humidity using a thermo-hygrostat. Filtered air was supplied using a pre-filter, medium filter, and HEPA filter. The management of the pigs and the facility and the welfare of the pigs were examined twice. Post approval monitoring was conducted once after the experiment.

### 2.4. Experimental Procedure

All animal experiments were conducted in Pusan National University Yangsan Hospital, Preclinical Trial Education Center from 9 April to 25 June 2019.

General anesthesia was induced with medetomidine hydrochloride, butorphanol tartrate, and midazolam and maintained with isoflurane with endotracheal intubation with the pigs in the horizontal supine position. A tracheostomy was performed to ventilate the left lung under bronchoscopic guidance. The ventilator was set to a tidal volume of 7–10 mL/kg, respiratory rate of 10 times/min, and 100% fraction of inspired oxygen.

A right thoracotomy was performed with the pigs in the left lateral position via the eighth intercostal space, and the eighth rib was cut from the ventral side because the intercostal space was narrow. The intercostal space was secured using a spreader; all procedures were performed with thoracoscopy. The thoracic duct was identified by dissecting around the descending aorta and the punctured site of the thoracic duct for lymphangiography by targeting the region as close as possible to the diaphragm. A 21-gauge needle was used to assess the thoracic duct, 0.018-inch guidewire was advanced through the needle into the thoracic duct, and a 3-French dilator with a stiffening cannula was inserted into the thoracic duct for lymphangiography and injecting iopamidol or lipiodol. Thoracic

duct damage was performed using a MEGADYNE™ E-Z CLEAN™ Needle 2.75-inch electrosurgical electrode (Ethicon US LCC, Cincinnati, OH, USA), with a distance of more than 5 cm from the tip of the cannula. Mono polar energy was delivered via an electrocautery with a cutting power of 10 W. The approximate size of the damage was a 1 mm diameter circle.

At the end of the experiment, the pigs under anesthesia were euthanized via vascular injection of KCl to alleviate animal suffering.

### 2.5. Study Design and Sample Size

Twelve pigs were equally and randomly divided into groups A and B. In group A, the pigs were injected with a solution of 0.5 mL indigo carmine and 2 mL iopamidol via the cannula into the thoracic duct immediately and at 5, 10, and 30 min after thoracic duct damage. During each injection, lymphangiography using iopamidol and video-assisted thoracoscopy was performed to monitor leakage. In group B, the pigs were injected with a solution of 0.5 mL indigo carmine and 2 mL lipiodol immediately and at 5, 10, and 30 min after the injury; lymphangiography and video-assisted thoracoscopy were performed to monitor chylous leakage after each injection. The rate of all injections was controlled using an infusion pump at 900 cc/h to avoid thoracic duct injury by flushing. Considering that it is feasible to make a direct comparison among products that are commonly used as contrast agents in clinical settings, we conducted a study using ISOVUE®-200 (Bracco Diagnostics Inc., Singen, Germany) Iopamidol Injection 41% and LIPIODOL® (Ethiodized Oil) (Guerbet, Villerpinte, France) Injection.

To assess the effects of iopamidol on tissue histology after the procedure, the thoracic ducts damaged during the experiment were collected from the pigs in group A and sent to the Department of Pathology for microscopic examination.

### 2.6. In Vitro Experiment

To determine the effects of mixing chyle with iopamidol or lipiodol in vitro, 1 mL of chyle was collected via the cannula from one pig in group B before thoracic duct damage and immediately dropped onto two slides. To one slide, 1 mL of iopamidol was added, and to the other, 1 mL of lipiodol was added, and then the samples were mixed. The slides were examined after 5, 10, and 30 min. Solidification was confirmed when a smear was not formed after gently touching the samples on the slides with filter paper. This is the Duke method, one of the methods for measuring bleeding time.

### 2.7. Statistical Analysis

Fisher's exact test was used to compare the positive proportion of pigs in each group at 5, 10, and 30 min. Statistical analyses were performed using IBM SPSS Statistics version 24 (IBM Corp., Armonk, NY, USA); results with $p < 0.05$ were considered significant.

## 3. Results

In group A, four of the six pigs did not exhibit chylous leakage as observed by lymphangiography performed 5 min after thoracic duct injury, and no additional leakage was observed after 10 and 30 min (Table 1). The remaining two pigs (Group A #1 and #4) showed more leakage for up to 5 min but not after 10 and 30 min. The results of the video-assisted thoracoscopy and lymphangiography of pig #4 in group A are shown in Figure 1. In group B, none of the pigs showed chylous leakage in the lymphangiography at 5, 10, or 30 min after thoracic duct injury. The results of the video-assisted thoracoscopy and lymphangiography of pig #4 in group B are shown in Figure 2. Nevertheless, the $p$ value was 0.46, and there was no statistically significant difference between groups A and B (Table 2).

**Table 1.** Comparison of chylous leakage by time zone in groups A and B.

| Group A | 1 | 2 | 3 | 4 | 5 | 6 |
|---|---|---|---|---|---|---|
| Immediately | + | + | + | + | + | + |
| 5 min | + | − | − | + | − | − |
| 10 min | − | − | − | − | − | − |
| 30 min | − | − | − | − | − | − |
| **Group B** | **1** | **2** | **3** | **4** | **5** | **6** |
| Immediately | + | + | + | + | + | + |
| 5 min | − | − | − | − | − | − |
| 10 min | − | − | − | − | − | − |
| 30 min | − | − | − | − | − | − |

+: leakage; −: blockage.

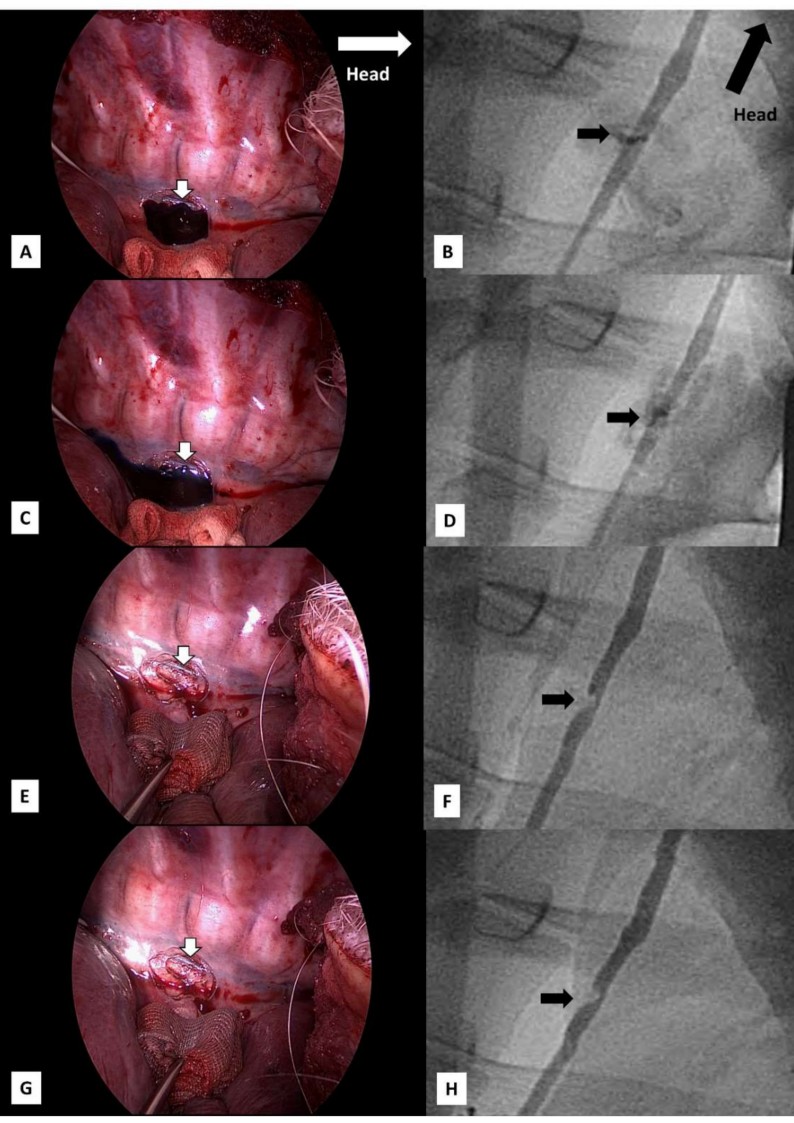

**Figure 1.** Leakage (arrow) shown by video-assisted (**A**) thoracoscopy and (**B**) lymphangiography immediately after thoracic duct injury by electrocautery, and (**C,D**) it was still evident 5 min after injecting 2 mL of iopamidol and 0.5 mL of indigo carmine into the distal thoracic duct. However, the leakage was healed at (**E,F**) 10 and (**G,H**) 30 min as shown in the surgical field and lymphangiography images. (Group A #4 pig).

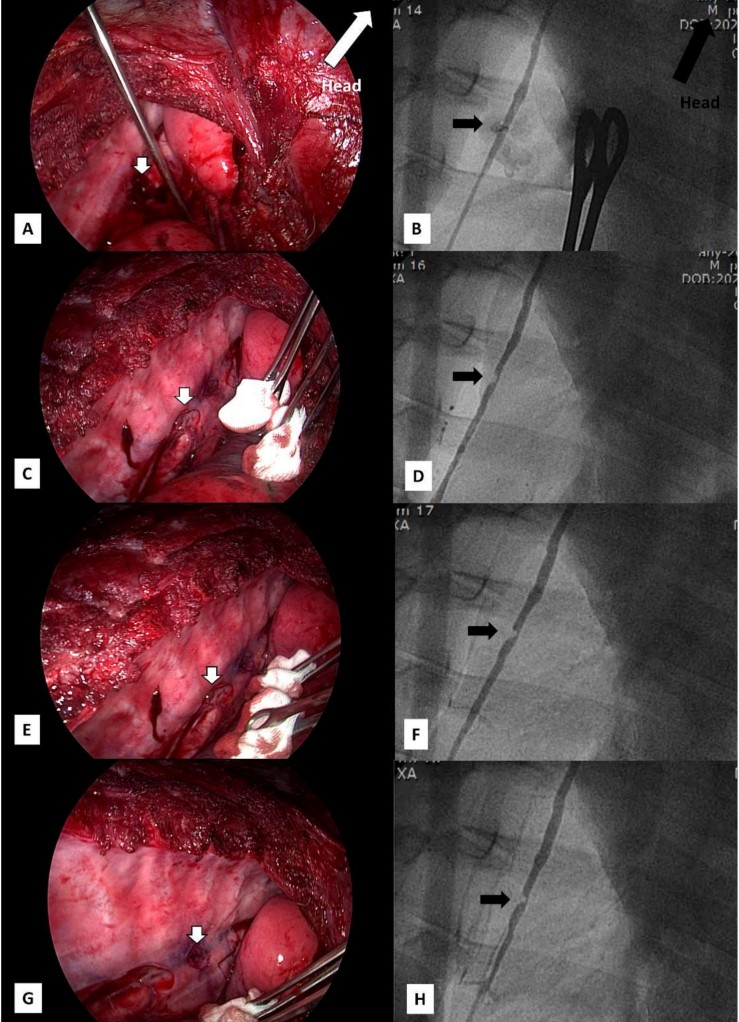

**Figure 2.** Leakage (arrow) observed by video-assisted (**A**) thoracoscopy and (**B**) lymphangiography immediately after thoracic duct injury by electrocautery, and the leakage was healed at (**C,D**) 5, (**E,F**) 10, and (**G,H**) 30 min as shown in the surgical field and lymphangiography images. (Group B #4 pig).

**Table 2.** Statistical analysis between groups A and B.

| Time (min) | | Group A | Group B | *p* |
|---|---|---|---|---|
| | | *n* = 6 | *n* = 6 | |
| 5 | + | 2 (33.3) | 0 (0.0) | 0.46 |
| | − | 4 (66.7) | 6 (100.0) | |
| 10 | + | 0 (0.0) | 0 (0.0) | 1 |
| | − | 6 (100.0) | 6 (100.0) | |
| 30 | + | 0 (0.0) | 0 (0.0) | 1 |
| | − | 6 (100.0) | 6 (100.0) | |

Fisher's exact test for categorical variables.

Figure 3 shows the results of the in vitro experiment performed by mixing pig (group B #1) chyle with iopamidol or lipiodol immediately and after 5 min; aggregation was observed in both mixtures, and the samples completely solidified at 30 min. The pathological observation of the damaged thoracic duct of the pigs after the experiment showed that the thoracic duct was filled with an amorphous proteinaceous material, indicating that iopamidol was retained at the injured site, followed by obstruction of the thoracic duct (Figure 4).

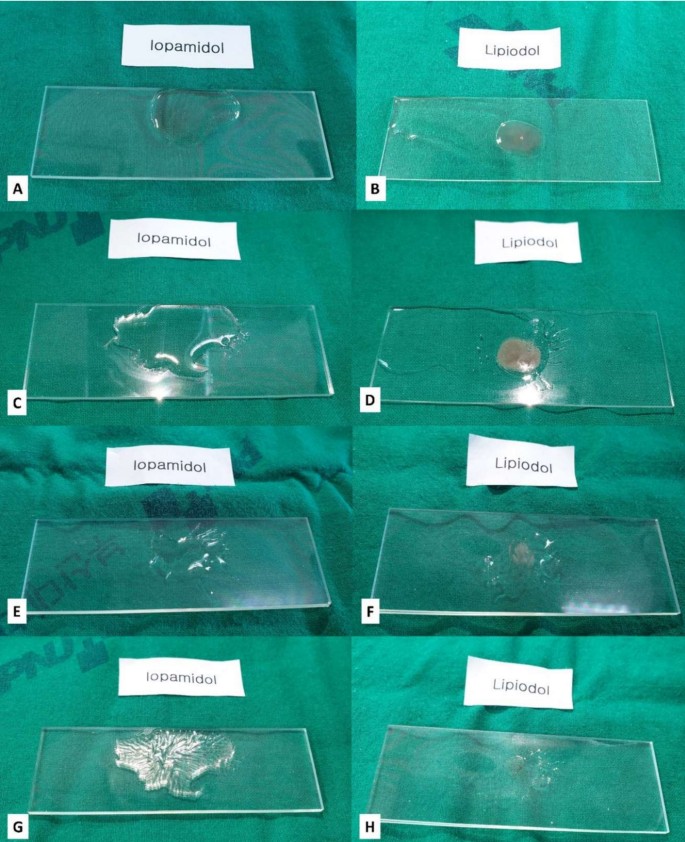

**Figure 3.** Image shows the conglomeration of chyle by iopamidol and lipiodol drops in vitro (on the slide) every 5, 10, and 30 min. (**A**,**B**) immediately and (**C**,**D**) 5, (**E**,**F**) 10, and (**G**,**H**) 30 min after dropping and mixing the samples on the slide. (Group B #1 pig).

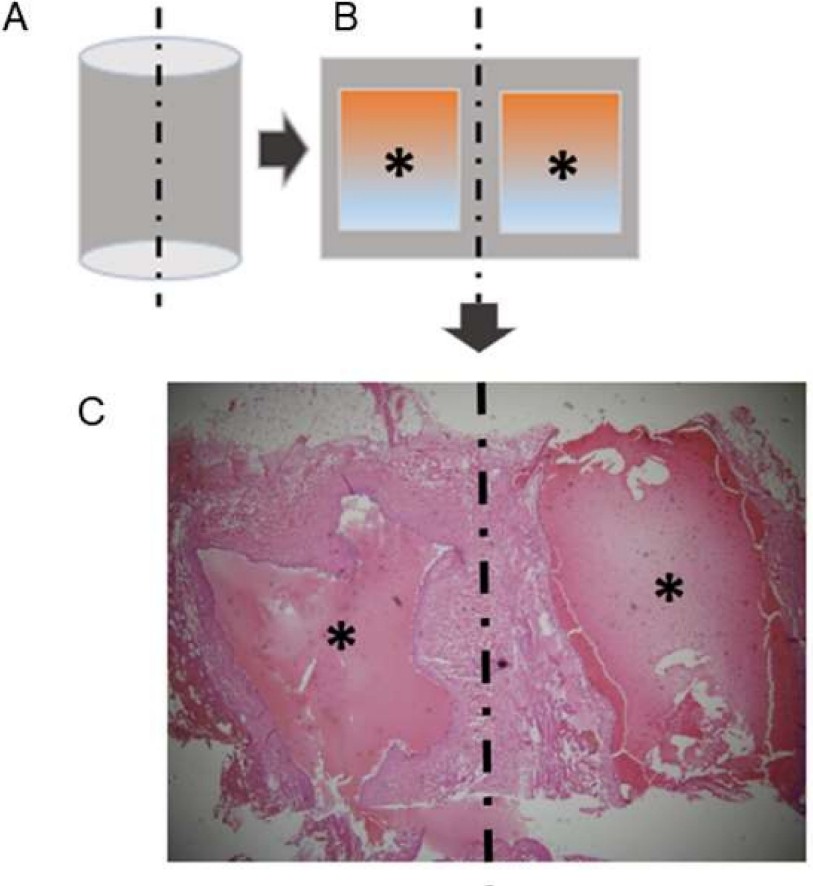

**Figure 4.** Pathological findings of a damaged thoracic duct. * Amorphous proteinaceous material (iopamidol) in the damaged thoracic duct. Amorphous proteinaceous material is stained red with hematoxylin and eosin stain, but it looks like a liquid rather than a cell. The thoracic duct is a long cylindrical tube; the image shows the collected cylindrical thoracic duct cut along the vertical plane and observed under a microscope. (**A**) represents the cylindrical shape of a thoracic duct and (**B**) represents the shape of a thoracic duct cut along the vertical plane. (**C**) The micrograph shows the left and right parts of a thoracic duct cut along the vertical plane. (hematoxylin and eosin stain, × 12.5) (Group A #2 pig).

## 4. Discussion

Post-operative chylothorax occurs in 0.5–2.0% of patients. Although most patients have been conservatively managed, treatments are often ineffective in patients with high flow leaks, requiring prolonged treatment [1–3,10]. Recent technological advances have enabled radiological interventions, which have shown relatively good clinical outcomes [5,6,9,11,13,14,19–22]. However, thoracic embolization or coilization may fail to occlude the thoracic duct in 30% of patients with post-operative chylothorax because of anatomical variations [6,13,20,23].

Despite the frequent failure of this procedure, chylous leakage has been shown to decrease immediately after lymphangiography, which by itself might have an important role in stopping chylous leakage [5,13,14,20,24,25]. This could be influenced by lipiodol, which is highly viscous and used as a therapeutic embolic agent. Owing to its characteristics, lipiodol is recommended for use with additional coils in an advanced or gelatin sponge to prevent non-target systemic embolization, such as in the brain and liver [26,27]. In addition, lipiodol should not be injected at doses exceeding the recommended limit, because it has the potential to cause systemic embolization.

We identified iopamidol, which can effectively seal the injured thoracic duct, as an alternative agent that avoids the adverse effects of lipiodol. Iopamidol is an iodinated, water-soluble, highly

viscous contrast agent that was able to conglomerate with chyle when dropped onto the slide (Figure 4). Furthermore, because of its water solubility [18], iopamidol would rarely cause embolization. The pharmacokinetics of iopamidol showed that the effects of iopamidol were the highest at 30–90 s after injection and that it was eliminated from the plasma in 4 days through the kidney, indicating that iopamidol does not persist longer than lipiodol [18,28].

Although a few studies have reported the use of lipiodol as a therapeutic embolic agent for lymphangiography [5,13–15,19], other agents, including contrast media such as iopamidol, have not been studied as potential options for this procedure. Our study showed that pin-point tears in the thoracic ducts injured by electrocautery during the operation were healed, and lymphangiography demonstrated prompt sealing of the leak at 10–30 min after the injection of iopamidol. As lipiodol can not only remain in the lymphatic system for a long time but also have a strong ability to embolize [29], the injury of the thoracic duct in all of the pigs might have recovered in just 5 min, unlike that in group A using iopamidol. However, owing to the strong ability of lipiodol, there is a risk of pulmonary embolization or cerebral embolization. Further studies using iopamidol are needed for safe thoracic duct embolization. The disappearance of the leak after iopamidol injection may be affected by the size of injury; therefore, more experiments with defects of varying sizes are needed to evaluate the efficacy of iopamidol.

To determine the mechanism of healing, the thoracic duct was extracted and sent to a pathologist for examination. The findings showed that the thoracic duct was full of amorphous proteinaceous material. This indicated that iopamidol was retained at the injured site, which led to the obstruction of the thoracic duct. The pathological findings revealed that iopamidol unpredictably filled and completely blocked the thoracic ducts. Although this is not observed in clinical practice, further studies are needed to determine whether the lumen of the thoracic duct is obstructed by iopamidol over time, as it is water-soluble. If the damaged area is sealed without leakage and the thoracic duct is communicated, iopamidol may be a better treatment option than TDE.

The FDA report on iopamidol [18] states that "Opacification of the calyces and pelves in patients with normal renal function becomes apparent within 1 to 3 min, with optimum contrast occurring between 5 and 15 min." Therefore, after the first injection of the contrast medium, we considered that it would take at least 5 min to avoid interference of the re-injected iopamidol with the previous injection. We decided to observe leakage at 5 and 10 min. Furthermore, as there may be re-leakage even if it was initially prevented, we decided to check leakage again at 30 min. When planning the experiment, we did not expect that iopamidol would prevent injury by filling the amorphous proteinaceous material within the thoracic duct, as observed in the histological analysis (Figure 4). Additionally, the lymphangiography findings presented in Figure 1F,H show that the damaged part (arrow) of the thoracic duct does not show contrast, as if an apple was hollowed inward. This further validates that some material filled the thoracic duct.

The use of iopamidol as a clinical therapeutic agent for lymphangiography has not been reported; our preliminary animal experiments showed that iopamidol may be an alternative therapeutic agent for treating chylous leakage. Additional animal studies and clinical trials to demonstrate the therapeutic effect of iopamidol on chylothorax may reduce or eliminate the incidence of systemic embolism induced by lipiodol. In addition, coil embolization may be avoided, which could reduce the treatment cost to and economic burden on the patient.

There were some limitations associated with this study because it was a pilot study to determine whether iopamidol, similar to lipiodol, can seal an injured thoracic duct. Firstly, there was a limitation in statistical analysis because the sample size was small and there was no sham group (pigs with thoracic duct injury and not treated with lipiodol or iopamidol) for comparison. Secondly, although the anatomical structure of the porcine lymphatic system is similar to that of humans, further research is needed to apply these findings directly to humans.

In conclusion, we showed the potential of iopamidol as a new therapeutic agent for chylous leakage in the present study. If thoracic duct embolization fails or if there is a risk of systemic embolization

following lipiodol injection, iopamidol injected via the thoracic duct may be a safe treatment that is worth further investigation.

**Author Contributions:** Conceptualization, H.Y.A., S.S., U.H., I.J.W., J.S.E., and D.-M.R.; investigation, H.Y.A., S.S., and U.H.; formal analysis, H.Y.A., S.S., and U.H.; resources, I.J.W. and J.S.E; project administration, D.-M.R.; writing—original draft preparation, H.Y.A., S.S., U.H., I.J.W., J.S.E., and D.-M.R.; writing—review and editing, H.Y.A., S.S., U.H., I.J.W., J.S.E., and D.-M.R. All authors have read and agreed to the published version of the manuscript.

**Funding:** This work was supported by the Biomedical Research Institute Grant, grant number 2019B022, Pusan National University Hospital.

**Conflicts of Interest:** The authors declare no conflict of interest.

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
