# Peer review of "Efficacy of Iopamidol for Sealing an Injured Thoracic Duct: Pilot Experiments in a Large Animal"

_applsci, doi:10.3390/app10238424_

Round 1

Reviewer 1 Report

Line 43: add further explanation or quantity "...systemic embolization may be an associated complication...". Recommend adding  a reference to this statement.

Line 47: add a reference.

Line 51: add a reference. 

General question: Why were the time points chosen (i.e. 5, 10, and 30 minutes)? If there is some clinical relevance, recommend including this in the manuscript, particularly the discussion section.

Section 2.6. In vitro experiment: Figure 2 states that mixing occurred after addition of iopamidol/lipiodol, but this is not stated or explained in the methods section. Include the mixing into the method section.

Section 2.6. In vitro experiment: Recommend removing 113 - 115 from "In vitro experiment, as it is not an In vitro experiment. Recommend creating another section or incorporating it into section 2.5 Study design and sample size.

Section 3 Results and/or Table 1: As it is stated in the method section (2.5) that leakage was observed for immediately after injury and administration of test articles, recommend adding a statement to confirm leakage was or was not observed at this time point. 

Section 3, line number 119: "...showed slight leakage..." is stated and it is not clear how this was determined, including after examining Figure 1. Recommend expanding further on how "slight leakage" was determined or replacing with a more appropriate descriptor. 

Figure 1 and line 128 - 132: As Lipiodol is used as a comparison, recommend adding images of Lipiodol treatment. Recommend adding Pig ID number into description in figure caption.

Line 134: recommend adding Pig ID number.

Figure 2 and line 141 - 143: add lipiodol into figure caption. currently there is no reference to it. 

Line 136: Explain how solidification was determined. Recommend adding this to the method section (section 2.6). If the degree of solidification differed between test articles (iopamidol and lipiodol), recommend discussing this in the discussion and/or results section.

Figure 3 and line 145 - 146: Add labels to figure 3 and/or further explanation in figure caption. It is unclear the purpose and value the top portions of the figure are providing. Recommend adding a label to where the healing (i.e. amorphous proteinaceous material) was observed. 

Line 158 and 160: recommend adding "non-target" before "systemic embolization". 

Line 164: add reference to "Furthermore, because of its water solubility, iopamidol would rarely cause embolization."

Line 165 - 167: Add reference to "Furthermore, because of its water solubility, iopamidol would rarely cause embolization. The pharmacokinetics of iopamidol showed that the effects of iopamidol were the highest at 30–90 s after injection and that it was eliminated from the plasma in 4 days through the kidney, indicating that iopamidol does not persist longer than lipiodol."

Line 176 - 177: Discuss and explain how "...full of amorphous proteinaceous material..." was determined. For example, was it through observation, use of stain or was there an additional identifying method?

Reviewer 2 Report

The study by Yeong Ahn et al is well designed, however the results are in a preliminar stage, since no statistical analysis is provided. I would recommend to:

1. Please explain in the introduction the side/adverse effects of lipiodol.

2. Since lipiodol can cause systemic embolization, I wonder whether they observed systemic embolization or how did they avoid/prevent systemic embolization in their model when using lipiodol.

3. In the methods section the volume of iopamidol used is mentioned but not the concentration, same for lipiodol. Did the authors used the same concentration for both compounds? how did they decide which concentration to use? did they performed a dose response curve? if so, please provide the data.

4. For the in vitro experiment is not clear to me why they only use chyle from group B and not from both A and B groups.

5. Figure 3 can be improved. I do not understand the differences between left and right histological images. Is it one treated with lipiodol and the other one wit iopamidol?.

6. Did the authors observed side effects when using lipiodol in their model? if so, please add this information in the manuscript.

7. Include statistical analysis. This will increase the impact of your findings.

Reviewer 3 Report

The manuscript explains the study and the outcome very well. I recommend following edits to the manuscript: 

1) This is a very preliminary pilot study with limited sample size without control group. This should be highlighted prominently in the manuscript. 

2) It is mentioned that the average age of the test subjects is 11 weeks. Why such a young population was selected for this study, and how does this impact the outcome of the rate of healing?

3) What was the size of the induced injury?

4) Please explain the reason for delayed effect of iopamidol compared to lipiodol.

Round 2

Reviewer 2 Report

Thank you to the authors for addressing all my comments. Since the statistical analysis does not show differences between groups A and B, probably due to the low n number, the conclusions and abstract of the study will need to be changed accordingly.

I would also recommend to add the concentrations of the two drugs in the methods section of the manuscript.

Author Response

We thank you for the valuable comment. We have added the following description in the Abstract (statistical analysis) and Method 2.5. Study design and sample size (concentrations).

Nevertheless, the p value was 0.46, and there was no statistically significant difference between groups A and B.

Considering that it is feasible to make a direct comparison among products that are commonly used as contrast agents in clinical settings, we conducted a study using ISOVUE®-200 Iopamidol Injection 41% and LIPIODOL® (Ethiodized Oil) Injection.

Reviewer 3 Report

Thank You for addressing the concerns in my earlier review.

Author Response

Thanks to your good opinion, we were able to make this paper even more complete.